# The Roles of miRNAs in Predicting Bladder Cancer Recurrence and Resistance to Treatment

**DOI:** 10.3390/ijms24020964

**Published:** 2023-01-04

**Authors:** Sanjna Das, Joshua Hayden, Travis Sullivan, Kimberly Rieger-Christ

**Affiliations:** 1Department of Translational Research, Lahey Hospital & Medical Center, Burlington, MA 01805, USA; 2Department of Urology, Lahey Hospital & Medical Center, Burlington, MA 01805, USA

**Keywords:** bladder cancer (BCa), recurrence, chemoresistance, miRNA, EMT, cell cycle, fatty acid metabolism, Wnt, HIPPO, FGFR3

## Abstract

Bladder cancer (BCa) is associated with significant morbidity, with development linked to environmental, lifestyle, and genetic causes. Recurrence presents a significant issue and is managed in the clinical setting with intravesical chemotherapy or immunotherapy. In order to address challenges such as a limited supply of BCG and identifying cases likely to recur, it would be advantageous to use molecular biomarkers to determine likelihood of recurrence and treatment response. Here, we review microRNAs (miRNAs) that have shown promise as predictors of BCa recurrence. MiRNAs are also discussed in the context of predicting resistance or susceptibility to BCa treatment.

## 1. Introduction

Bladder cancer (BCa) is among the most common human malignancies, resulting in over 17,000 deaths annually in the United States [1]. Prognosis of BCa is largely dependent on whether there is muscle invasion present or not. Approximately 75% of bladder tumors are non-muscle invasive at the time of diagnosis, and even with treatment approximately 40% will recur and 10% will progress. [2]. Muscle-invasive bladder cancer (MIBC) is associated with greater morbidity and mortality, and is typically managed more aggressively than non-muscle invasive bladder cancer (NMIBC), often requiring neoadjuvant chemotherapy, radical cystectomy, and pelvic lymph node dissection.

Progression to muscle-invasive disease is a critical element to address in reducing the morbidity and mortality associated with BCa and may be dependent on several factors. First, BCas have historically been described as arising from two distinct pathways: a non-invasive pathway characterized by mutations in oncogenes, and an invasive pathway characterized by mutations in tumor suppressor genes [3]. Specific genetic alterations may indeed influence risk of developing muscle invasive disease. Recent work has evaluated differential RNA expression in BCa in order to characterize the disease into distinct molecular subtypes, which links genetic alterations with clinical characteristics such as recurrence and progression of disease [4,5,6,7,8]. Mutations in genes that impact chromosomal stability, such as *DDR*, *P53* and *APOBEC* genes are associated with high risk of recurrence and progression, while alterations in genes involved in early cell-cycle processes such as *RAS*, *FGFR3*, and uroplakin genes are associated with low risk of recurrence and progression [9,10].

Several studies have identified microRNAs (miRNAs) as potential biomarkers for diagnosing and predicting survival among patients with BCa [11,12,13]. MiRNAs are short, noncoding molecules that negatively regulate gene expression by binding to the untranslated regions of gene transcripts. In addition to miRNAs, there are long non-coding RNAs (lncRNAs) and circular RNAs (cirRNAs); these are both classes of non-coding RNAs that work within axes involving miRNAs to regulate gene expression and that will be discussed briefly. MiRNAs are readily isolated from cell-free matrices such as urine [14] and serum [15] of patients with BCa with minimal risk to the patient. In BCa, they have been studied in conjunction with the epithelial to mesenchymal transition (EMT) phenotype [16,17], as well as chemoresistance and recurrence. The focus of this manuscript are those miRNAs that have been specifically linked to BCa resistance and recurrence. Several recent studies that have examined miRNA-based biomarkers and identified the roles of these miRNAs in BCa recurrence and resistance; this has the potential to inform the management of BCa patients.

## 2. Methods

In this review, PubMed (https://www.ncbi.nlm.nih.gov/pmc/) and Google searches from 19 September through 24 October 2022 were used to identify miRNAs associated with either BCa recurrence or chemoresistance. Search terms used to identify relevant papers include microRNA, bladder cancer, chemoresistance, recurrence, and selection focused on primary research papers that also included gene targets. A selection of papers looking at other non-coding RNAs were also included to highlight that (i) they also play a role in bladder cancer outcomes and that (ii) axes involving miRNAs frequently also include non-coding RNAs. These miRNAs and details pertaining to their potential gene targets are listed in Table 1 and Table 2 below. MiRNAs identified in the search were then analyzed in KEGG pathway analysis using DIANA-mirPath v. 3.0 [18]. Predicted gene targets are based on the experimentally validated miRNA interactions derived from TarBase v. 7.0 [19] (University of Thessaly, Volos, Greece).

The resulting heat maps depict significant pathways resulting from pathways union analysis using FDR correction and conservative statistics with a modified Fisher’s Exact Test, *p* < 0.0001. The results of the KEGG analyses are included below in Figure 1 and Figure 2. Notable pathways for recurrence include cell cycle, transcriptional regulation in cancer, the Hippo signaling pathway, and fatty acid metabolism/biosynthesis. For chemoresistance, these pathways include TGF-beta, fatty acid metabolism/biosynthesis, cell cycle, and the Hippo signaling pathway. These pathways and their aberrant expression are linked to cancer-associated phenotypes such as EMT.

## 3. EMT

An important mechanism linked to both recurrence and chemoresistance is EMT. This process involves the conversion of epithelial to mesenchymal cells resulting from changes in polarity and adhesion [67], and is characterized at the molecular level by changes including E-cadherin downregulation and N-cadherin upregulation [67]. Notably, EMT is a mechanism of metastasis, since the aforementioned molecular, and consequently cellular, changes allow cancer cells to move to other areas of the body using means such as migration and invasion [68]. There is a significant body of work investigating the role of EMT in BCa, especially with respect to recurrence and chemoresistance, the central themes of this review. Specifically, studies have looked at EMT-related genes to predict prognostic outcomes in patients with BCa [50,69] and at the role of non-coding RNAs that regulate the EMT pathway through axes involving miRNAs [70,71]. One important group of miRNAs involved in BCa EMT is the miR-200 family: work has shown that miR-200c and miR-200b inhibit EMT [72,73], and miR-200 expression is broadly linked to improved bladder cancer survival [74]. Other miRNAs involved in the EMT phenotype and included in Table 1 include miR-302b, which inhibits proliferation, migration, and invasion [34] and let-7f-5p, which is also characterized by tumor suppressor activity and inhibits cell viability and migration [35]. In addition to recurrence, EMT is also involved in mediating chemoresistance; in fact, EMT, recurrence, and chemoresistance work in concert with one another. Studies have found that EMT markers are upregulated in chemoresistant cells, with restoration of sensitivity via TGF-beta downregulation [75] and that oncogenic proteins modulate EMT to induce resistance to chemotherapeutic agents [76,77]. Two key pathways that have been linked to the EMT phenotype are Wnt signaling [78,79] and TGF-beta [80,81]. While the TGF-beta pathway is not focused on in detail here, there are several manuscripts looking at non-coding RNAs, including miRNAs, that are involved in BCa carcinogenesis (and specifically EMT): This includes miR-758-3p which is regulated by the lncRNA CASC9 [82], miR-663, which represses invasion and migration [83], miR-143-3p, which acts as a tumor suppressor and whose downregulation via LINC02470 promotes EMT [84], and miR-200b, which inhibits the metastatic phenotype [85]. TGF-beta signaling has also been linked to chemoresistance in a pathway that involves miR-145 downregulation [86]. The Wnt pathway, on the other hand, will be explored in greater detail in connection to PTEN, a protein target regulated by miR-21 (Table 2) and whose downregulation has been linked to poor prognosis and chemoresistance in BCa.

## 4. Cell Cycle

The cell cycle, or the process through which cells grow, undergo DNA replication, and ultimately divide is carefully regulated, and disruptions can result in adverse consequences. The cycle is separated into five phases: G1, and G2, which precede DNA replication (S) and the mitotic (M) phases, respectively, and G0, a dormant phase [87]. One key family of proteins that regulates the cycle is cyclin-dependent kinases (CDKs), which work in concert with cyclin protein substrates [87]. Cyclins and their associated kinases have been implicated in BCa, where they show biomarker and treatment potential [88,89]. While there does not appear to be any work showing that miRNAs directly bind to and regulate the expression of cyclins in connection to BCa recurrence, work by Wu et al. 2022 described an axis in which circGLIS3 sponges miR-1273f, effectively inhibiting its expression [90]. This leads to expression of SKP1 and cyclin D1, promoting BCa cell proliferation [90]. This is an excellent example of how different classes of noncoding RNAs such as cirRNAs and miRNAs work together to regulate gene expression and consequently the cellular phenotype. Additionally, miR-138, included in Table 1, is positively correlated with cyclin D3 expression [23].

Cyclin D3, and FGFR3, a well-studied protein explored below, may be used to identify BCa patients who are likely to recur using noninvasive methods [91]. Additional work has explored the utility of cyclins as biomarkers for recurrence including in papillary urothelial bladder cancer [92,93]. There is very limited literature looking at cyclins in BCa chemoresistance, and the potential role of non-coding RNAs; this represents an area requiring further exploration.

## 5. FGFR3

The Fibroblast growth factor receptor 3 (FGFR3) gene is often mutated in BCa, and these mutations may be associated with less aggressive cancers and better patient outcomes [94,95,96]. FGFR3 is included here because of its importance in BCa development, the progress that has been made so far in developing FGFR3-focused treatments, and due to its potential regulation by several miRNAs in BCa [97], including miR-99/100, which may be connected to recurrence. Measuring miR-100 levels in either blood or urine may offer insight into FGFR3 expression and aid in identifying patients who may benefit from targeted therapies [96]. More broadly, there is a plethora of information around FGFR3 in recurrence. Recent work by Sikic et al. 2021 found that FGFR3 expression was correlated with a higher likelihood of recurrence [98], and an earlier study looked at FGFR3 as one of three urine biomarkers for BCa recurrence [99]. Studies of FGFR3 in BCa are fairly well developed; in fact, there are clinical trials examining FGFR3 inhibitors in cancer treatment. Two such trials are PROOF 302 (phase III), which is investigating the utility of infigratinib, an inhibitor, in a specific subset of BCa patients [100] and FIGHT-101 (phase I/II), which is evaluating the treatment potential of pemigatinib, another FGFR inhibitor, in patients across different cancers including bladder [101]. Interestingly, while the PROOF 302 patient population includes only patients with FGFR3 mutations, FIGHT-101 is nonspecific [100,101].

There is more limited information around FGFR3 in chemoresistance although FGFR3 mutations have been linked to chemoresistance via Akt pathway activation [102,103]. On the flip side, FGFR3 mutations in conjunction with ERCC1 expression has also been reported to confer chemosensitivity [104]. This suggests that response to chemotherapy in patients with FGFR3 mutations varies according to the specific genetic alteration. Interestingly, di Martino et al. 2019 describe overlap between FGFR3 and the Hippo pathway, discussed further below. They found that FGFR3 acts through ETV5 and ultimately TAZ upregulation to initiate morphological changes promoting BCa progression [105]. Convergence between FGFR3 and the Hippo pathway is an important finding, because TAZ may represent an important target to halt BCa progression.

## 6. Hippo Signaling

One pathway implicated in the KEGG analyses for both BCa recurrence and chemoresistance is Hippo. This pathway is involved in regulating organ size and is characterized as a tumor suppressor, with its dysregulation linked extensively to cancer [106,107,108]. One of the key molecules within the pathway is the oncogenic Yes-associated protein (YAP) [106], which works in concert with TAZ and TEAD to control proliferation and apoptosis at the transcriptional level [109]. Proliferation and apoptosis are characteristics whose regulation (or lack thereof) is linked to invasive potential and consequently recurrence. While there is limited work examining the role of YAP in BCa recurrence, a study by Ghasemi et al. reported higher YAP expression in recurrent BCa [110]. There is additional work looking more broadly at agents that modulate BCa progression through YAP [111,112] and at molecules, including miRNAs, that affect BCa development through YAP [113,114]. YAP is also involved in BCa chemoresistance, with studies examining the naturally-derived ailanthone as an inhibitor of proliferation and migration in cisplatin-resistant BCa cells through YAP and Nrf2 repression [115,116]. Nrf2 is another transcription factor regulating phenotypic characteristics including chemoresistance, and it appears to communicate with YAP in modulating response to chemotherapy, with work showing that repression of the two increases chemosensitivity [117]. YAP, described in Table 1, along with Nrf2, are targets that merit further exploration in BCa treatment and have been explored by Cheng et al. where they discuss YAP-targeting agents as a way to address chemoresistance [118].

## 7. Wnt Signaling

Another important mechanism is the Wnt signaling pathway, which plays a role in organismal development [119]. Aberrant Wnt signaling is observed in BCa, especially in relation to invasive characteristics and EMT [120]. Specifically, studies have identified molecules such as EFEMP2, PYCR1, and TMEM88 which work through Wnt signaling to promote BCa invasiveness [78,121,122]. There is significantly less information around the connection between the Wnt pathway and BCa recurrence. A study by Cai et al. 2022 identified several lncRNAs associated with BCa prognosis in which KEGG analyses implicated Wnt among the associated signaling pathways [123]. There is limited work on miRNAs and Wnt signaling in recurrence, an area that warrants further investigation, as it could yield additional targets for BCa therapy. Non-coding RNAs, including miR-148b-3p, noted in Table 2, also regulate Wnt signaling, affecting the response of BCa cells to chemotherapy [124]. PTEN, a binding target of miR-148b-3p, exerts an anticancer effect in BCa cells via Wnt downregulation [124]. In addition to PTEN, another molecule in the Wnt pathway that is often modulated is beta-catenin, a transcription factor [119] whose downregulation results in greater chemosensitivity of BCa cells [125].

## 8. Fatty Acid Metabolism and Synthesis

In the DIANA miRPATH KEGG analyses for BCa recurrence and chemoresistance, both fatty acid biosynthesis and metabolism were implicated. Fatty acids represent an energy source for cancer cells, which obtain these acids through processes including lipogenesis and fatty acid intake [126]. Jeong et al. 2021 found that upregulation of proteins that mediate fatty acid intake is associated with poorer pathological and clinical outcomes [126]. In addition to these proteins, Abdelrahman et al. 2019 reported that upregulation of fatty acid synthase (FASN), as well as E2F1 and Her2/neu expression, is associated with BCa recurrence [127,128]. FASN has been reported to act through AKT and CCND1 to promote survival and growth in BCa [128]. In addition to the proteins implicated in fatty acid metabolism in BCa, downstream molecules like AKT and CCND1 may hold treatment and/or biomarker potential. More limited work addresses the role of fatty acids in chemoresistance. Okamura et al. 2021 found that miR-486-5p, downregulated in cisplatin-resistant BCa cell lines, binds to EHHADH, which has been implicated in fatty acid metabolism [129]. Additionally EHHADH is involved in metastatic characteristics including migration and invasion [129], and therefore may also mediate resistance.

## 9. Conclusions

Analysis of miRNAs, along with other non-coding RNAs, can guide cancer treatment and management by offering insight into, for example, tumors with a higher likelihood of recurrence and that may be resistant to certain chemotherapeutic agents. It is evident from this review that there is a need for additional work at the cellular level to explore associations that have been reported between miRNAs and recurrence. Specifically, there is a paucity of work exploring (i) the effects of miRNAs implicated in recurrence on the invasive phenotype and (ii) targets, regulators, and pathways through which these miRNAs may act. Identifying these molecules can help in developing novel treatments, as is the case with FGFR3. There is also a need for more work to identify molecules regulating and regulated by miRNAs that are implicated in response to chemotherapy. Once gene targets implicated in recurrence and treatment response are identified, the pathways in which they are involved should be further explored, since these may aid in bladder cancer treatment. Pathways that are important for additional investigation include Hippo, Wnt, and fatty acid metabolism, and promising gene targets include YAP and PTEN, in addition to the well-researched FGFR3.

Though this review focuses on miRNAs, there are other non-coding RNAs, such as long non-coding RNAs and circular RNAs, that have similarly been implicated in BCa resistance and recurrence and which should be explored further, especially since these networks also involved miRNAs [51,130,131,132]. This includes Cdr1as, a circular RNA that mediates bladder cancer chemotherapeutic response via miR-1270 [132] and circ-BPTF, which promotes recurrence through interaction with miR-31-5p [133]. Likewise, the circular non-coding RNA serum biomarkers circFARSA, circSHKBP1, and circBANP were found to be able to discriminate patients with recurrent BCa [134]. In addition, the long non-coding RNA FOXD2-AS1 was found to promote bladder cancer progression and recurrence through a feedback loop with Akt [135]. In a study of the TCGA BCa dataset, Zhang et al. [136] identified a biomarker signature composed of 14 long non-coding RNA to predict recurrence free survival in BCa. These studies illustrate the roles, and potential utility, of other non-coding RNA in aiding BCa treatment.

Ultimately, studies looking at miRNAs, their gene targets and associated pathways, and even other non-coding RNAs can serve as biomarkers for recurrence and resistance. This information can be used to develop cell-free assays (involving either urine or serum/plasma) and offer simple, noninvasive methods to identify patients likely to recur and those who may exhibit chemoresistance to specific drugs. This is clinically relevant as a key component of BCa management centers around prevention of progression of NMIBC to muscle-invasive disease. It is important to note that there is existing work exploring liquid biopsies as a tool in bladder cancer recurrence: this includes Urovysion, Xpert, and more preliminary findings looking at miRNAs that may be useful [15,137,138]. In addition, there are clinical trials investigating the potential of biomarkers to aid in treatment decisions. Recently, a clinical trial conducted by the SWOG Cancer Research Network reported on a gene-expression biomarker called the COXEN GC score [139]. This test aims to predict tumor response to drug treatment. Likewise, the active BISCAY trial (NCT02546661), is attempting to use protein and tumor antibody biomarkers in an effort to identify patients that are likely to repond to treatment. Also, the TOMBOLA trial (NCT04138628) measures circulating tumor DNA in an effort to identify early metastasis and thus aid in initiating early immunotherapy. Collectively these works illustrate the coming age of biomarkers aiding in clinical decision-making.

Management strategies for NMIBC such as intravesical instillation of bacillus Calmette-Guérin (BCG), intravesical chemotherapies, and even systemic immunotherapies such as pembrolizumab play an important role in reducing risk of progression to muscle-invasive disease, but are dependent on individuals’ response to these therapies, which can vary based on patient- and tumor-level factors. Determining a patient’s response to treatment presents a significant challenge and although progress has been made, currently there are no molecular biomarkers used in the clinical setting to predict response to these therapies [140].

Tumor-specific factors that can affect progression from NMIBC to MIBC include variant histology, which is defined as BCa with histology other than typical urothelial carcinoma (i.e., micropapillary, nested, plasmacytoid, neuroendocrine, or sarcomatoid tumors). NMIBCs with variant histology are associated with increased risk of upstaging to muscle-invasive disease, tend to respond less to minimally invasive treatment modalities such as intravesical instillation of BCG, and are more frequently managed with guideline-directed yet highly morbid procedures such as radical cystectomy when compared to typical urothelial carcinomas [141,142]. Interestingly, work has been done to understand how surgical margin further affects survival following radical cystectomy [143]. Though it is understood that NMIBC with variant histology represents higher risk disease [144], certain variant histologic subtypes carry higher risk than others, and debate exists on how tumor histology should guide the aggressiveness of treatment (i.e., intravesical therapies vs. cystectomy). Moreover, identification of variant histology is dependent on pathologists’ interpretation and therefore intrinsically subjective. As such, adjunctive tests may provide additional predictive value to the physician making treatment decisions. Other biomarker-based tools in bladder cancer management include Controlling Nutritional Status (CONUT) [145] and molecules such as PD-L1 [146]. MiRNAs offer important information pertaining to molecules and signaling pathways that can be regulated to effectively manage bladder cancer in conjunction with additional tools. Little is known about which miRNAs may serve as predictors of variant histology and subsequent disease progression among patients who have tumors with variant histology [12,147]. Identifying such miRNAs may aid physicians in making treatment decisions for the management of non-muscle invasive BCa.

In summary, further understanding of miRNAs and their potential to serve as biomarkers for identification of high-risk patients who may respond poorly to conservative therapies, or whose pathways may lead to the development of targeted therapies that can prevent the morbidity associated with radical cystectomy, may contribute to improved patient outcomes and decreased morbidity associated with the management of BCa.

## Figures and Tables

**Figure 1 ijms-24-00964-f001:**
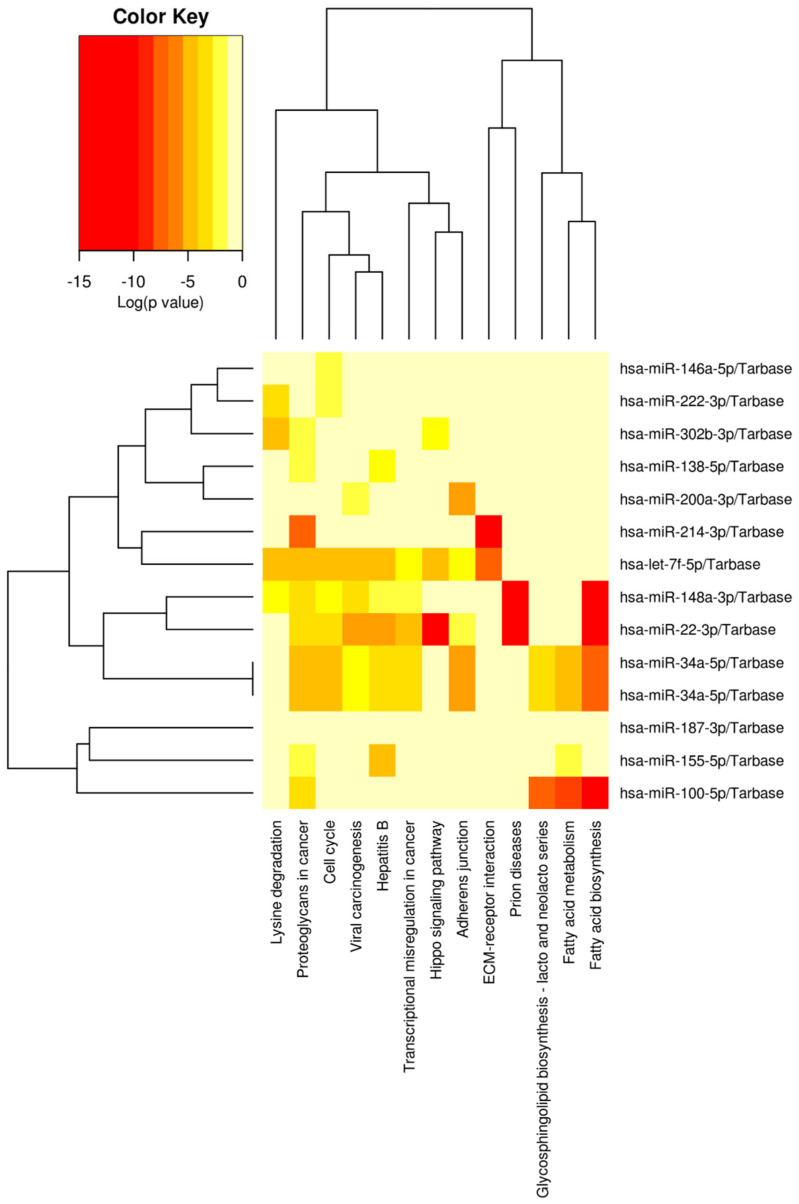
DIANA miRPath [18] KEGG analysis results identified several pathways that may be related to recurrence in bladder cancer. Relevant pathways, based on predicted gene targets of these miRNAs, include proteoglycans in cancer (92 genes), Hippo signaling (49 genes), cell cycle (69 genes), adherens junctions (32 genes), ECM receptor-interaction (15 genes), fatty acid metabolism (14 genes), and fatty acid biosynthesis (4 genes).

**Figure 2 ijms-24-00964-f002:**
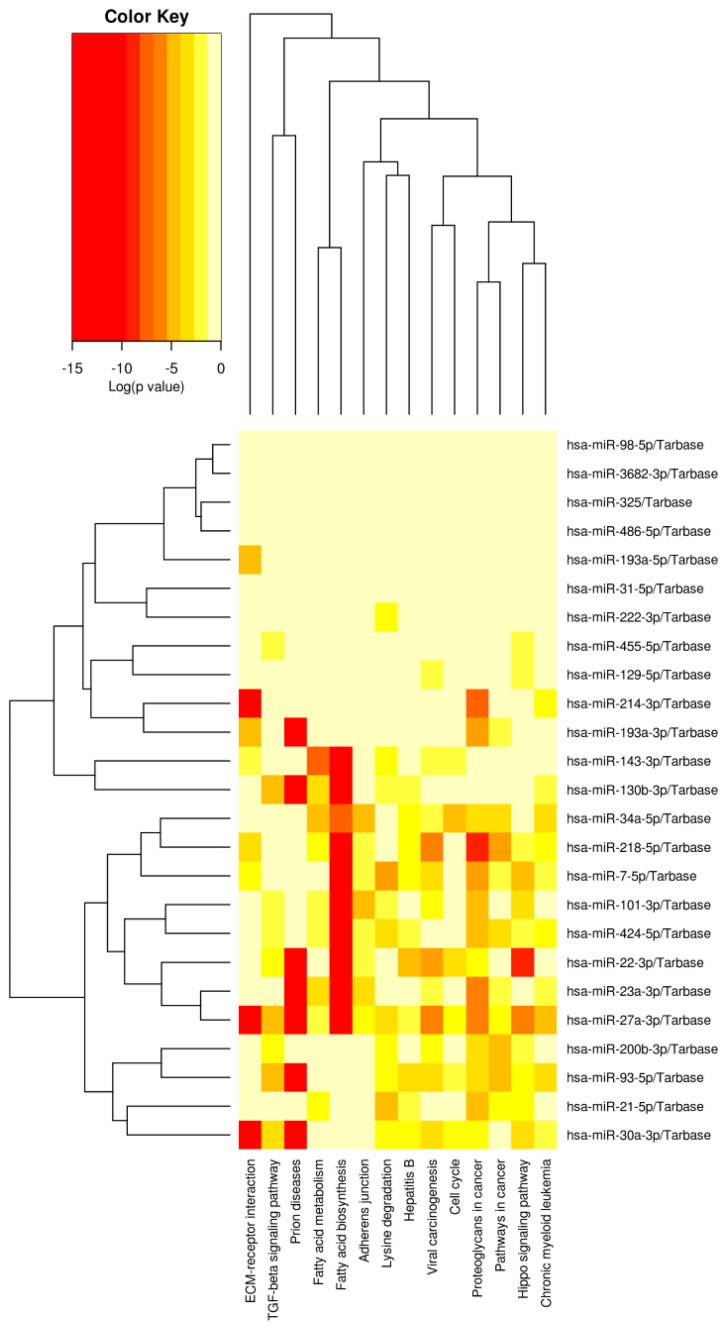
DIANA miRPath KEGG analysis results identified several pathways that may be related to chemoresistance in BCa. Relevant pathways, based on predicted gene targets of these miRNAs, include proteoglycans in cancer (124 genes), Hippo signaling (68 genes), cell cycle (52 genes), TGF-beta signaling (44 genes), adherens junctions (40 genes), ECM receptor-interaction (31 genes), fatty acid metabolism (15 genes), and fatty acid biosynthesis (6 genes).

**Table 1 ijms-24-00964-t001:** miRNAs and associated targets/regulators involved in BCa recurrence.

miRNA	Target/Regulator	Function	Reference
MiR-22-3p	Not identified	Elevated miR-22-3p showed reduced recurrence-free survival (RFS).	[20]
MiR-34a	Not identified	Downregulation associated with recurrence and poorer prognosis [21]. Higher expression of miR-34a associated with lower likelihood of recurrence. MiR-34a upregulation showed less invasion and colony formation [22].	[21,22]
MiR-100	FGFR3	Reduced miR-100 associated with less recurrence. *	[23]
MiR-138	Cyclin D3	Downregulation of miR-138 linked to recurrence.	[23]
MiR-146a-5p	Two separate pathways involving YAP1 and COX2	Downregulation associated with recurrence. Subsequent regulation of ALDH1A1 and SOX2.	[24]
MiR-148a	Not identified	Downregulation of miR-148a in BC patients linked to recurrence and metastasis.	[25]
MiR-152	Not identified	Higher expression of miR-152 linked to lower RFS in NMIBC.	[26]
MiR-155	Not identified	MiR-155 upregulation associated with recurrence.	[27]
MiR-187-5p	Not identified	Oncogene, promotes proliferation and mobility while decreasing apoptosis.	[28]
MiR-200a family	Not identified	Reduced miR-200a-3p showed reduced RFS [20]. Lower expression of miR-200a in BCa, and downregulation linked to higher chance of recurrence [29].	[20,29]
MiR-210	Not identified	Higher expression of miR-210 found in patients with recurrence.	[30]
MiR-214	Not identified	Reduced miR-214 expression in BCa urines pre-op compared to post. Linked to RFS [31]. Mir-214 downregulation linked to recurrence [32].	[29,30,31,32]
MiR-221/222	Not identified	Downregulated in BCa, but miR-222 is upregulated in high grade/invasive BCas. MiR-222 upregulation (Ta/T1 cancers) linked to recurrence.	[33]
MiR-302b	EPS8 (potential)	Tumor suppressor, lessens proliferation, migration, and invasion. Promotes apoptosis.	[34]
Let-7f-5p	LIN28	Tumor suppressor, represses cell viability and migration.	[35]

* *p* value > 0.05.

**Table 2 ijms-24-00964-t002:** miRNAs and associated targets/regulators involved in BCa chemoresistance.

miRNA	Target/Regulator	Function	Reference
MiR-7-5p	ATG7	Upregulation of miR-7-5p inhibited invasive characteristics. Promotes chemosensitivity.	[36]
MiR-21	PTEN	Promotes chemoresistance to doxorubicin and proliferation in transitional cell carcinoma; inhibits doxorubicin-induced apoptosis.	[37]
MiR-22-3p	NET1	MiR-22-3p promotes chemoresistance. More cell viability, colony formation, and less apoptosis with upregulation of miR-22-3p via mimic.	[38]
MiR-23a	SFRP1 protein and Wnt signaling	Linked to chemoradiation response.	[39]
MiR-27a	SFRP1 protein and Wnt signaling, RUNX-1	Linked to chemoradiation response [39]. Rs11671784 SNP (wherein A is replaced with G) results in reduced chemosensitivity [40].	[39,40]
MiR-30a-3p	MMP2, MMP9	Combination of cisplatin and miR-30a-3p resulted in improved apoptosis and reduced cell viability. Upregulation of miR-30a-3p via mimic lessened migration and invasion.	[41]
MiR-31	ITGA5	MiR-31 promotes chemosensitivity to mitomycin-C and upregulation inhibits proliferation, migration, and invasion. Downregulation associated with higher risk of recurrence in noninvasive UBC.	[42]
MiR-34a	TCF1, LEF1, Cdk6, SRT-1 (sirtuin), CD44	Downregulated in BCa; promotes chemosensitivity to epirubicin [43] and to cisplatin [44,45]. Higher expression of miR-34a represses metastatic characteristics [43,45].	[43,44,45]
MiR-93	LASS2 (but no direct binding)	MiR-93 promotes chemoresistance.	[46]
MiR-98	LASS2	Expressed at higher levels in BCa. Upregulation via mimic resulted in increased proliferation, greater cisplatin and doxorubicin resistance, and repression of apoptosis.	[47]
MiR-101	COX2	MiR-101 promotes chemosensitivity to cisplatin.	[48]
MiR-101-3p	EZH2, affects MRP1 expression	MiR-101-3p promotes chemosensitivity.	[49]
MiR-129-5p	Wnt5a	Expression of miR-129-5p promotes response to gemcitabine.	[50]
MiR-130b	CYLD	Involved in promoting chemoresistance.	[51]
MiR-143	IGF-1R	MiR-143 promotes chemosensitivity. Upregulation of IGF-1R linked to reduced survival and recurrence.	[52]
MiR-193a-3p	LOXL4, HOXC9, PSEN1, ING5	MiR-193a-3p promotes chemoresistance (oxidative stress pathway) [53,54]. MiR-193a-3p reported to target PSEN1 gene and affect DNA damage response [55]. Interaction with ING5 also occurs through DNA damage response pathway [56].	[53,54,55,56]
MiR-193a-5p	AL-2α	MiR-193a-5p is involved in chemoresistance. Upregulation of miR-193a-5p linked to increased migration and resistance to cisplatin.	[57]
MiR-200b	IGFBP3, ICAM1, TNFSD10	MiR-200b promotes chemosensitivity. More broadly, miR-200 family members (miR-200b, miR-200a, and miR-429) were downregulated in cisplatin-resistant cell lines.	[58]
MiR-214	Netrin-1	Tumor suppressor activity; miR-214 upregulation resulted in reduced colony formation and invasion. MiR-214 promotes chemosensitivity.	[59]
MiR-218	Glut1	MiR-218 promotes chemosensitivity to cisplatin.	[60]
MiR-222	PPP2R2A	MiR-222 is implicated in chemoresistance. Acts through AKT/mTOR and autophagy pathways.	[61]
MiR-325	HAX-1	MiR-325 promotes chemosensitivity.	[62]
MiR-424	UNC5B and SIRT4	Promotes cisplatin resistance via downregulation of UNC5B and SIRT4.	[63]
MiR-455-5p	Regulated by HOXA-As3	Promotes sensitivity to cisplatin, reduces proliferation, and promotes apoptosis.	[64]
MiR-486-5p	Gene expression changes observed in caspase-9, caspase03, P53, SIRT1, OLFM4, SMAD2, Bcl-2, ROCK, CD44, MMP9	MiR-486-5p functions as tumor suppressor and promotes chemosensitivity.	[65]
miR-3682-3p	Regulated by BMI1 and regulates ABCB1	BMI1 inhibits miR-3682-3p transcription to induce chemoresistance. Elevated BMI1 is also associated with poorer RFS.	[66]

## Data Availability

Available upon request.

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
