# Peer review of "The Roles of miRNAs in Predicting Bladder Cancer Recurrence and Resistance to Treatment"

_ijms, 2023, doi:10.3390/ijms24020964_

Round 1
Reviewer 1 Report
The authors have well described the roles of miRNAs involved in predicting recurrence and treatment resistance in bladder cancer.
However, I think it is unreasonable to introduce miRNAs in bladder cancer and refer to them as potential biomarkers based only on the results of a somewhat pubmed search.
A great number of papers have been published on miRNAs in bladder cancer over the years, and I think there is a lack of explanation as to why only the miRNAs introduced in this review were selected among them. Rather, it would be better to select and introduce miRNAs that are more closely related to recurrence predictive markers and treatment resistance markers.
In addition, recently, many studies on the possibility of cancer diagnosis and recurrence prediction through miRNA-mRNA-long noncoding RNA network studies including miRNA have been conducted. I hope to have more comments on this area.
Finally, to predict the recurrence of bladder cancer, a diagnostic test through liquid biopsy is required, and research is focused worldwide.
In order for this paper to provide useful information to the reader, we hope to further introduce the role of miRNAs in bladder cancer based on liquid biopsy.
Author Response
Reviewer 1
Comments and Suggestions for Authors
The authors have well described the roles of miRNAs involved in predicting recurrence and treatment resistance in bladder cancer. However, I think it is unreasonable to introduce miRNAs in bladder cancer and refer to them as potential biomarkers based only on the results of a somewhat pubmed search. A great number of papers have been published on miRNAs in bladder cancer over the years, and I think there is a lack of explanation as to why only the miRNAs introduced in this review were selected among them. Rather, it would be better to select and introduce miRNAs that are more closely related to recurrence predictive markers and treatment resistance markers.
Thank you for your comment. We have added a Methods section to clarify the focus of this review along with additional citations. The focus of this review is indeed miRNA directly related to recurrence and resistance. We have strengthened our terminology in an effort to emphasize this.
In addition, recently, many studies on the possibility of cancer diagnosis and recurrence prediction through miRNA-mRNA-long noncoding RNA network studies including miRNA have been conducted. I hope to have more comments on this area.
Thank you for your suggestion. We have added a paragraph to the Discussion pertaining to these topics.
Finally, to predict the recurrence of bladder cancer, a diagnostic test through liquid biopsy is required, and research is focused worldwide. In order for this paper to provide useful information to the reader, we hope to further introduce the role of miRNAs in bladder cancer based on liquid biopsy.
Thank you for your suggestion. We have added several studies that look at liquid biopsies in bladder cancer recurrence.
Reviewer 2 Report
In this review the authors sought out to clarify the role of miRNAs in predicting BCa recurrence and resistance to treatment. This is an intriguing field of research among uro - oncological literature and the authors should be commended for their efforts.
Following some comments.
Please add a "Methods" section in which you can further describe the research strategy (time and date of research, only Pubmed-based search?)
As these novel emerging tools are not ready-to-use in the clinical daily setting the authors would consider a brief introduction about the role of conventional pathological (variant histologies - doi: 10.1007/s00345-020-03364-z, positive surgical margins - doi: 10.1007/s00345-021-03776-5), immunohistochemical (FGFR3 mutation status, p53 and KI-67 expression evaluation, PD-1, PD-L1) and clinical-laboratoriostic (inflammation-, inflammation-nutrion-based biomarkers: NLR, albumin to globulin ratio, albumin to fibrinigen ratio, PNI, CONUT and others) markers available to highlight the current knowledge and gaps. How miRNAs can integrate these available tools? Are these ready for the prime time or for widespread diffusion?
In this specific context, please further refer to these recent manuscripts about microRNAs across different settings of BCa treatment (doi: 10.4111/icu.2016.57.S1.S60, doi: 10.1016/j.ajur.2021.05.001, doi: 10.1097/MD.0000000000022891, doi: 10.1002/cam4.1570, doi: 10.1002/1878-0261.12523)
It would be interesting to discuss what are the current clinical trials on the topic of biomarker-selected cohorts (for example NCT04138628 IMvigor 010). There are any expected trials in this specific setting using miRNAs?
Author Response
Reviewer 2
In this review the authors sought out to clarify the role of miRNAs in predicting BCa recurrence and resistance to treatment. This is an intriguing field of research among uro - oncological literature and the authors should be commended for their efforts.
Following some comments.
Please add a "Methods" section in which you can further describe the research strategy (time and date of research, only Pubmed-based search?)
Thank you for your comment. We have added the Methods section as suggested.
As these novel emerging tools are not ready-to-use in the clinical daily setting the authors would consider a brief introduction about the role of conventional pathological (variant histologies - doi: 10.1007/s00345-020-03364-z, positive surgical margins - doi: 10.1007/s00345-021-03776-5), immunohistochemical (FGFR3 mutation status, p53 and KI-67 expression evaluation, PD-1, PD-L1) and clinical-laboratoriostic (inflammation-, inflammation-nutrion-based biomarkers: NLR, albumin to globulin ratio, albumin to fibrinigen ratio, PNI, CONUT and others) markers available to highlight the current knowledge and gaps. How miRNAs can integrate these available tools? Are these ready for the prime time or for widespread diffusion?
Thank you for your suggestion. The above studies have been added along with information pertaining to the role of miRNAs in bladder cancer management.
In this specific context, please further refer to these recent manuscripts about microRNAs across different settings of BCa treatment (doi: 10.4111/icu.2016.57.S1.S60, doi: 10.1016/j.ajur.2021.05.001, doi: 10.1097/MD.0000000000022891, doi: 10.1002/cam4.1570, doi: 10.1002/1878-0261.12523)
Thank you for your suggestion. The papers listed above have been added to the manuscript.
It would be interesting to discuss what are the current clinical trials on the topic of biomarker-selected cohorts (for example NCT04138628 IMvigor 010). There are any expected trials in this specific setting using miRNAs? https://pubmed.ncbi.nlm.nih.gov/31481066/ recurrence paper with circRNA https://pubmed.ncbi.nlm.nih.gov/30103209/
We appreciate you identifying these studies. We have included them, along with several other clinical trials utilizing biomarkers in aiding BCa clinical decision making. In addition, we have included the circRNA paper you referenced in our section detailing other non-coding RNAs linked to bladder cancer recurrence.
Round 2
Reviewer 1 Report
The author did a good job fixing most of the requests.
I believe that this paper can provide readers with a variety of information because it has been significantly revised.
Reviewer 2 Report
The authors revised the manuscript properly. Congratulations.